# Kinetochore-fiber lengths are maintained locally but coordinated globally by poles in the mammalian spindle

**Manuela Richter[1,2]\*, Lila Neahring[2,3], Jinghui Tao[2], Renaldo Sutanto[2†], Nathan H Cho[1,2], Sophie Dumont[1,2,3,4,5]\***

[1]Tetrad Graduate Program, University of California, San Francisco, San Francisco, United States; [2]Department of Bioengineering & Therapeutic Sciences, University of California, San Francisco, San Francisco, United States; [3]Developmental & Stem Cell Biology Graduate Program, University of California, San Francisco, San Francisco, United States; [4]Biochemistry & Biophysics Deptartment, University of California, San Francisco, San Francisco, United States; [5]Chan Zuckerberg Biohub, San Francisco, United States

**Abstract** At each cell division, nanometer-scale components self-organize to build a micron-scale spindle. In mammalian spindles, microtubule bundles called kinetochore-fibers attach to chromosomes and focus into spindle poles. Despite evidence suggesting that poles can set spindle length, their role remains poorly understood. In fact, many species do not have spindle poles. Here, we probe the pole's contribution to mammalian spindle length, dynamics, and function by inhibiting dynein to generate spindles whose kinetochore-fibers do not focus into poles, yet maintain a metaphase steady-state length. We find that unfocused kinetochore-fibers have a mean length indistinguishable from control, but a broader length distribution, and reduced length coordination between sisters and neighbors. Further, we show that unfocused kinetochore-fibers, like control, can grow back to their steady-state length if acutely shortened by drug treatment or laser ablation: they recover their length by tuning their end dynamics, albeit slower due to their reduced baseline dynamics. Thus, kinetochore-fiber dynamics are regulated by their length, not just pole-focusing forces. Finally, we show that spindles with unfocused kinetochore-fibers can segregate chromosomes but fail to correctly do so. We propose that mammalian spindle length emerges locally from individual k-fibers while spindle poles globally coordinate k-fibers across space and time.

## Editor's evaluation

The authors find that the control of overall kinetochore fiber length in mitotic spindles of cultured mammalian cells does not require spindle pole focusing. Pole focusing is however required for fine-tuning their length and for correct chromosome segregation. Technically sophisticated experiments and careful quantitative data analysis provide compelling evidence for the drawn conclusions that provide valuable insight into spindle morphogenesis. This work is of particular interest to cell biologists and biophysicists interested in cytoskeleton organization.

## Introduction

Living systems use simple, small-scale components to build larger and more complex structures. One such structure is the micron-scale spindle, built from nanometer-scale tubulin molecules. The length of the spindle dictates the distance over which chromosomes segregate in dividing cells, and

**\*For correspondence:**
manuela.richter@ucsf.edu (MR);
sophie.dumont@ucsf.edu (SD)

**Present address:** †Cell & Developmental Biology Graduate Program, University of Michigan, Ann Arbor, United States

**Competing interest:** The authors declare that no competing interests exist.

spindle length is known to scale with cell size during development (*Good et al., 2013*; *Hazel et al., 2013*; *Lacroix et al., 2018*; *Rieckhoff et al., 2020*; *Wühr et al., 2008*). Defects in spindle length are linked to impaired chromosome segregation (*Goshima et al., 1999*), cytokinesis errors (*Dechant and Glotzer, 2003*), and asymmetric division defects (*Dudka et al., 2019*; *Dumont et al., 2007*), and long spindles have been hypothesized to come at an energetic cost (*Dumont and Mitchison, 2009a*). While we know many proteins that can modulate the spindle's length (*Goshima and Scholey, 2010*), how they work together to set spindle length and ensure robust chromosome segregation remains poorly understood. We do not know which aspects of spindle length and dynamics are regulated by global cues at the level of the whole spindle, and which are more locally regulated at the level of its components.

Mammalian spindles are built from a network of microtubules, including discrete bundles of microtubules connecting chromosomes to poles. These bundles, called kinetochore-fibers (k-fibers), are made of many microtubules, some of which directly extend from kinetochores to poles (*Kiewisz et al., 2022*; *McDonald et al., 1992*; *O'Toole et al., 2020*). Poles are the convergence points of k-fiber microtubules and other microtubule minus-ends, and they can also serve as an anchor point for centrosomes, if present, and astral microtubules. In many systems, dynein and other motors work together to focus microtubules into asters and poles (*Compton, 1998*; *Goshima et al., 2005a*; *Heald et al., 1996*; *Merdes et al., 1996*; *Roostalu et al., 2018*; *So et al., 2022*). In mammals, k-fiber microtubules turn over on the order of minutes (*Gorbsky and Borisy, 1989*), detaching from kinetochores and getting replaced. They also exhibit poleward flux, where k-fiber tubulin moves toward poles, with k-fiber plus-ends on average polymerizing and minus-ends appearing to depolymerize at poles (*Mitchison, 1989*). Both biochemical factors (*Goshima and Scholey, 2010*) and mechanical force (*Akiyoshi et al., 2010*; *Dumont and Mitchison, 2009b*; *Nicklas and Staehly, 1967*) are thought to tune k-fiber dynamics at both microtubule ends and thereby tune k-fiber length. Microtubule dynamics regulators with length-dependent activities (*Dudka et al., 2019*; *Mayr et al., 2007*; *Stumpff et al., 2008*; *Varga et al., 2006*) could in principle give rise to the k-fiber's length scale, beyond simply tuning length. However, k-fiber architecture and organization vary across species, adding complexity to our understanding of how k-fibers set their length. Some spindles, such as in land plants, do not have focused poles (*Yamada and Goshima, 2017*), and in many species, spindles are composed of short, tiled microtubules indirectly connecting chromosomes to poles (*Brugués et al., 2012*; *Yang et al., 2007*), unlike mammalian k-fibers. Broadly, it remains poorly understood which of the mammalian spindle's emergent properties—such as length, dynamics, and function—emerge globally from the whole spindle, or locally from individual k-fibers themselves.

While we know that perturbations that affect spindle pole-to-pole distance also affect k-fiber length, and vice versa (*Waters et al., 1996*), it is still unclear which sets the other. For example, global forces such as cell confinement pulls on poles, leading to k-fiber elongation by transiently suppressing apparent minus-end depolymerization (*Dumont and Mitchison, 2009b*)**,** but pole-less k-fibers do not elongate under these forces (*Guild et al., 2017*). Similarly, locally pulling on a k-fiber with a microneedle causes it to stop depolymerizing at its pole and thus elongate (*Long et al., 2020*). Since poles serve as a connection point for spindle body microtubules, centrosomes, and astral microtubules, they can in principle help integrate physical and molecular information from within and outside the spindle. Indeed, one proposed model is that force integration at spindle poles sets mammalian k-fiber length and dynamics (*Dumont and Mitchison, 2009a*).However, focused poles may not be essential for setting spindle length, as species without focused poles (*Yamada and Goshima, 2017*) can still build spindles and set their length. Similarly, inhibiting dynein unfocuses poles but spindles still form albeit with altered lengths in *Drosophila* (*Goshima et al., 2005b*) and *Xenopus* (*Gaetz and Kapoor, 2004*; *Heald et al., 1996*; *Merdes et al., 1996*), and without a clear effect on mammalian spindle length (*Guild et al., 2017*; *Howell et al., 2001*). Further, it is possible to alter kinetochores and microtubule dynamics to shorten k-fibers without a corresponding decrease in the spindle's apparent length (*DeLuca et al., 2006*). The role of the mammalian spindle pole on k-fiber structure, dynamics, and function remains an open question.

Here, we ask which emergent properties of mammalian k-fibers require a focused spindle pole. We inhibit pole-focusing forces and ask how k-fiber length, dynamics, and function change when the spindle reaches an unfocused steady state. Using live imaging, we find that k-fibers can set their mean length without poles but need poles to homogenize and coordinate their lengths between k-fibers. To

test whether unfocused k-fibers can recover their lengths, as control ones do, we acutely shorten them using laser ablation or a microtubule-destabilizing drug and show that they recover their length. They do so by tuning their end dynamics and recover more slowly due to reduced baseline dynamics. Thus, k-fiber length is not simply regulated by global pole-focusing forces, but by local length-based mechanisms. Lastly, we show that while the mammalian spindle can move chromosomes without focused poles, it does so with severe segregation and cytokinesis defects. Together, this work indicates that mammalian spindle poles and pole-focusing forces are not required for k-fiber length establishment and maintenance, but for coordinating spindle structure, dynamics, and function across space and time. We propose that the spindle length scale emerges locally at the level of an individual k-fiber, and that robust, coordinated spindle architecture and function arise globally through spindle poles.

## Results

### Spindle poles coordinate but do not maintain kinetochore-fiber lengths

To test whether k-fiber length is set locally or globally, we generated metaphase spindles without focused poles, but with a steady-state length at metaphase. To do so, we overexpressed the dynactin subunit p50 (dynamitin) in PtK2 mammalian rat kangaroo cells, a system with few chromosomes and clearly resolved individual k-fibers. p50 overexpression dissociates the dynactin complex and inhibits the pole-focusing forces of its binding partner, dynein (*Echeverri et al., 1996*; *Howell et al., 2001*; *Quintyne et al., 1999*), unfocusing poles in species such as *Xenopus* and *Drosophila* (*Gaetz and Kapoor, 2004*; *Sharp et al., 2000*). Indeed, we found that unfocused spindles correlated with higher mean intensity levels of p50 expression (*Figure 1—figure supplement 1*), consistent with prior work showing mild pole disruption vs. severe unfocusing depending on the severity of dynein inhibition (*Elting et al., 2017*; *Gaglio et al., 1997*, p. 199; *Hueschen et al., 2019*; *Hueschen et al., 2017*; *Sharp et al., 2000*, p. 200; *van Toorn et al., 2022*). To probe the role of poles on global and local spindle architecture, here we selected unfocused spindles that maintained a steady-state structure on the minutes timescale, holding their shape over time (*Figure 1—video 3*).

We first imaged unfocused spindle assembly in cells overexpressing p50 using long-term confocal fluorescence live imaging with a wide field of view to capture these rare events. While k-fibers seemed initially focused in these cells, these k-fibers eventually lost their connection to centrosomes and became unfocused, exhibiting a similar phenotype to spindle assembly in some NuMA-disrupted cells (*Figure 1A*, *Figure 1—videos 1 and 2*, *Silk et al., 2009*). We observed disconnected centrosomes seemingly move around freely in cells with unfocused spindles (*Figure 1—videos 2 and 3*). The resulting metaphase spindles were barrel-shaped with bi-oriented chromosomes, and they underwent anaphase after several hours instead of about 30 min in control, consistent with dynein inhibition at kinetochores causing an anaphase delay (*Howell et al., 2001*; *Figure 1A*, *Figure 1—videos 1 and 2*). While these spindles had no clear poles, we sometimes observed transient clustering of neighboring k-fibers, likely due to residual pole-focusing forces from other minus-end motors or incomplete dynein inhibition. Their interkinetochore distance was indistinguishable from control (*Figure 1—figure supplement 2*), suggesting that k-fibers are still under some tension from other forces (*Elting et al., 2017*; *Kajtez et al., 2016*; *Maiato et al., 2004*; *Milas and Tolić, 2016*), despite not being connected to poles. p50 overexpression in human RPE1 cells led to similar unfocusing phenotypes (*Figure 1—figure supplement 3*), but k-fibers were not individually resolvable. Thus, we chose to work with p50 overexpression in PtK2 spindles and hereafter refer to these spindles and k-fibers without distinct poles and with reduced pole-focusing forces as 'unfocused'.

To measure k-fiber lengths more accurately, we imaged control and unfocused spindles at metaphase using short-term confocal fluorescence live imaging at higher spatial resolution (*Figure 1B*). If poles do not contribute to k-fiber length, we expect no change in k-fiber length distributions in unfocused spindles (*Figure 1Ci*). If poles are required to set spindle length, we expect k-fibers with a different mean length in unfocused spindles (*Figure 1Cii*). If poles merely coordinate lengths, we expect k-fibers with a greater variability of lengths in p50 spindles, but the same mean length (*Figure 1Ciii*). We first observed that in unfocused spindles, k-fibers were more spread out in the cell, with spindles covering a larger area compared to control along both its major and minor axes (*Figure 1D*, *Figure 1—figure supplement 4A*). This is consistent with pole-focusing forces providing contractile forces to compact the spindle (*Hueschen et al., 2019*). Next, we measured k-fiber lengths

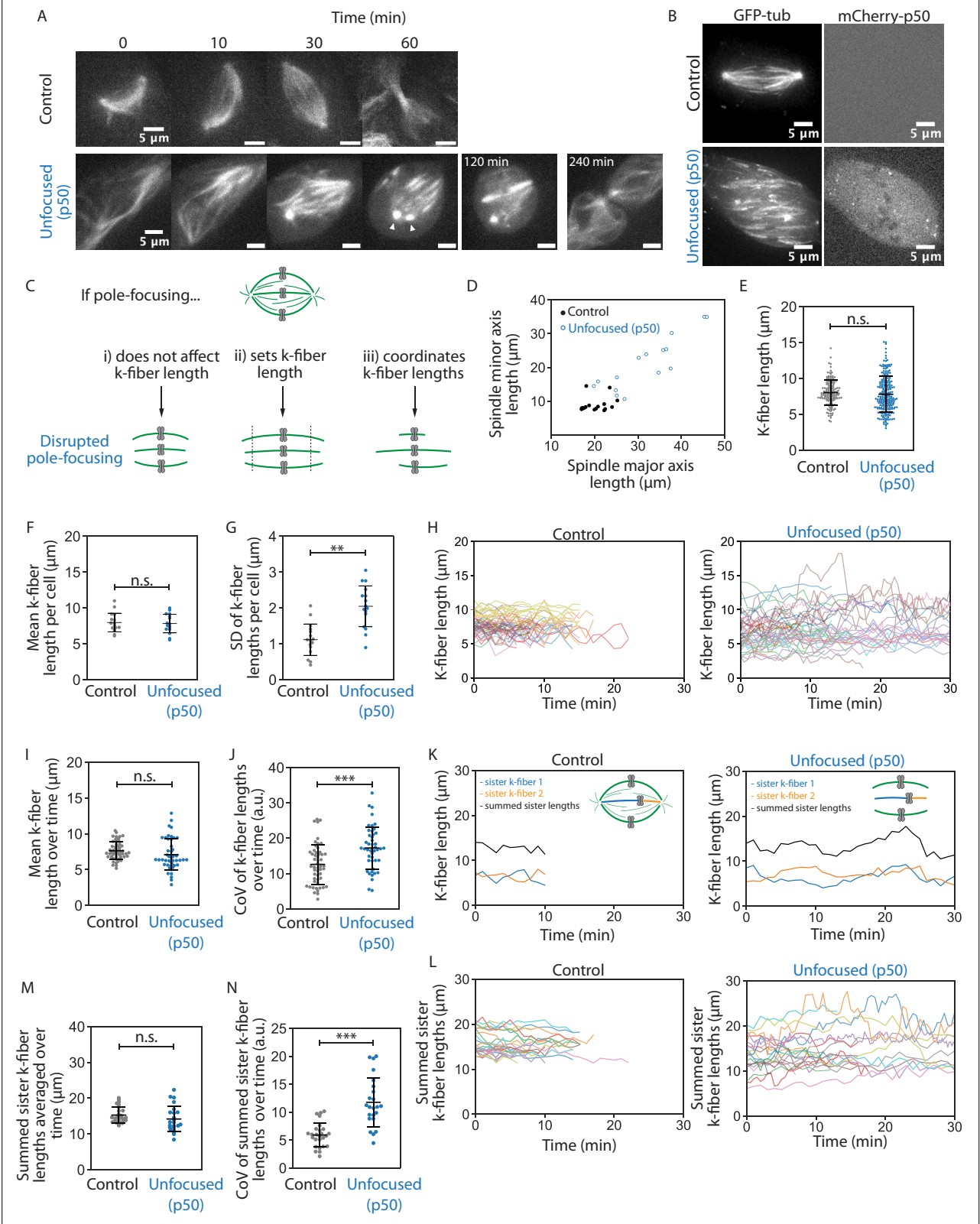

**Figure 1.** Spindle poles coordinate but do not maintain kinetochore-fiber lengths. See also *Figure 1—videos 1–3*. (**A**) Representative confocal timelapse images of spindle assembly showing max-intensity z-projections of HaloTag-β-tubulin PtK2 spindles labeled with JF 646, from nuclear envelope breakdown at *t* = 0 through cytokinesis. mCherry-p50 was infected into unfocused but not control cells. Arrowheads mark where both centrosomes were observed to be disconnected from the spindle. (**B**) Max-intensity z-projections of representative confocal images of PtK2 spindles

*Figure 1 continued on next page*

*Figure 1 continued*

with GFP-α-tubulin (control and unfocused) and mCherry-p50 (unfocused only). (**C**) Cartoon model of a mammalian spindle with chromosomes (gray) and microtubules (green), with predictions for k-fiber lengths after disrupting poles. Figures D–G are from the same dataset (Control: *N* = 16 cells; Unfocused: *N* = 16 cells). (**D**) Spindle major and minor axis lengths in control and unfocused spindles (major axis: Control = 20.24 ± 2.65 μm, Unfocused = 31.87 ± 7.85 μm, p = 6.3e−5; minor axis: Control = 8.96 ± 2.12 μm, Unfocused = 21.23 ± 7.61 μm; p = 2.5e−5; Control *N* = 16, Unfocused *N* = 15). (**E**) Lengths of control and unfocused k-fibers from z-stacks by live-cell imaging (Control: *n* = 144 k-fibers, 8.01 ± 1.76 μm; Unfocused: *n* = 222 k-fibers, 7.81 ± 2.52 μm; p = 0.38). (**F**) Mean lengths of control and unfocused k-fibers averaged by cell (Control: 7.97 ± 1.30 μm; Unfocused: 7.84 ± 1.31 μm; p = 0.79). (**G**) Length standard deviation of control and unfocused k-fibers per cell (Control: 1.12 ± 0.44 μm; Unfocused: 2.05 ± 0.58 μm; p = 2.9e−5). Figures H–N are from the same dataset (Control: *N* = 9 cells, *n* = 52 k-fibers; Unfocused: *N* = 9 cells, *n* = 46 k-fibers). (**H**) Lengths of k-fibers measured over time in control and unfocused spindles. Each trace represents one k-fiber; each color represents a cell. (**I**) K-fiber length averaged over time in control and unfocused spindles. Each point represents one k-fiber (Control: 7.64 ± 1.23 μm; Unfocused: 7.09 ± 2.19 μm; p = 0.14). (**J**) Coefficients of variation for k-fiber lengths over time in control and unfocused spindles. Each point represents one k-fiber (Control: 12.60 ± 5.62 a.u.; Unfocused: 17.23 ± 5.98 a.u.; p = 1.8e−4). Figures K–N were analyzed by sister k-fiber pairs (Control: *N* = 9 cells, *n* = 26 k-fiber pairs; Unfocused: *N* = 9 cells, *n* = 23 k-fiber pairs). (**K**) Lengths of sister k-fibers were measured over time in control and unfocused spindles. One representative k-fiber for each condition is shown in orange, its sister in blue, and their sum in black. (**L**) The sum of sister k-fiber lengths over time in control and unfocused spindles. Each trace is one sister k-fiber pair. (**M**) Summed sister k-fiber lengths averaged over time (from L). Each dot represents one sister k-fiber pair (Control: 15.27 ± 2.19 a.u.; Unfocused: 14.18 ± 3.54 a.u.; p = 0.22). (**N**) Coefficient of variation of summed sister k-fiber lengths over time (from L). Each dot represents one sister k-fiber pair (Control: 5.90 ± 2.14 μm; Unfocused: 11.77 ± 4.34 μm; p = 2.4e−6). Numbers are mean ± standard deviation. Significance values determined by Welch's two-tailed *t*-test denoted by n.s. for p ≥ 0.05, * for p < 0.05, ** for p < 0.005, and *** for p < 0.0005.

The online version of this article includes the following video, source data, source code, and figure supplement(s) for figure 1:

**Source code 1.** This script thresholds spindle images and fits an ellipse to calculate major and minor axis lengths.

**Source data 1.** This spreadsheet contains the data used to generate plots 1D, 1E, 1F, 1G, S2, S3, S4, and S5.

**Source data 2.** This spreadsheet contains the data used to generate plots 1H, 1I, 1J, 1K, 1M, 1N, and 3A.

**Figure supplement 1.** High cytoplasmic p50 intensity correlates with unfocused spindles.

**Figure supplement 2.** Interkinetochore distance is preserved in unfocused spindles.

**Figure supplement 3.** p50 overexpression in RPE1 cells generates unfocused spindles.

**Figure supplement 4.** Length measurement methods.

**Figure supplement 5.** Centrosome radius approximation.

**Figure supplement 6.** Kinetochore-fiber lengths are spatially correlated in control but not unfocused spindles.

**Figure 1—video 1.** Control spindle assembly in the presence of pole-focusing forces.
https://elifesciences.org/articles/85208/figures#fig1video1

**Figure 1—video 2.** Spindle assembly with inhibited pole-focusing forces.
https://elifesciences.org/articles/85208/figures#fig1video2

**Figure 1—video 3.** Kinetochore-fiber lengths over time in metaphase: control vs. unfocused spindle.
https://elifesciences.org/articles/85208/figures#fig1video3

in 3D (*Figure 1—figure supplement 4B, C*). For control spindles whose k-fibers end at centrosomes at this resolution, we subtracted the radius of the centrosome (0.97 ± 0.10 μm) from the region of measured tubulin intensity (*Figure 1—figure supplement 5*). Mean k-fiber length in an unfocused spindle (7.81 ± 2.52 μm) was not significantly different than control (8.01 ± 1.76 μm) (*Figure 1E*). Thus, k-fibers do not require a pole connection to keep their mean length. However, these unfocused spindles showed a greater standard deviation in lengths, so we compared average k-fiber lengths per cell to account for cell-to-cell variability: the mean k-fiber length within each cell was indistinguishable between control and unfocused cells (*Figure 1F*), but the standard deviation was significantly greater in unfocused cells (*Figure 1G*). This indicates that spindle poles act to synchronize lengths between neighbors within a spindle, rather than to set and keep length. K-fibers can maintain their average length without poles, but they do so with a greater length variability.

In principle, this greater k-fiber length variability in unfocused spindles could not only come from greater length variability between k-fibers in a given cell (*Figure 1G*), but also from greater variability over time for each k-fiber. To test this idea, we measured k-fiber lengths over time (*Figure 1H*, *Figure 1—video 3*). We observed indistinguishable mean lengths averaged over time in unfocused and control k-fibers and a greater coefficient of variation in unfocused k-fiber lengths over time compared to control (*Figure 1I, J*). Thus, while unfocused k-fibers still establish and maintain their

mean lengths at a similar length scale (*Figure 1F, I*), their lengths are more variable within a cell (*Figure 1G*) and over time (*Figure 1J*) compared to control.

To test the role of poles in coordinating lengths within the spindle across space, we tested whether k-fiber length correlated with k-fiber spatial positioning within the spindle. Based on geometry and previous observations, we expected outer k-fibers to be longer than inner k-fibers in focused spindles (*Kiewisz et al., 2022*). This was indeed the case in control spindles, but this difference was lost in unfocused spindles (*Figure 1—figure supplement 6A*). Furthermore, we expected k-fiber length to correlate with distance to the metaphase plate – k-fibers are shorter if attached to under-aligned chromosomes and longer if attached to over-aligned chromosomes (*Wan et al., 2012*). Here too, a correlation between k-fiber length and alignment was observed in control but it was negligible in unfocused spindles (*Figure 1—figure supplement 6B*). Thus, poles coordinate k-fiber lengths spatially in the spindle to maintain its shape despite geometric constraints.

Finally, to test the role of poles in coordinating lengths within the spindle across time, we compared sister k-fiber lengths over several minutes. During chromosome oscillations, sister k-fiber lengths are normally anti-correlated (*Wan et al., 2012*). Indeed, in control cells we observed that as one sister k-fiber shortened, the other elongated to maintain a constant sum of their lengths. However, this was not observed in unfocused spindles (*Figure 1K*). In unfocused spindles, the sum of sister k-fiber lengths was indistinguishable from control when averaged over time, but their sum was less conserved over time, yielding higher coefficients of variation (*Figure 1K–N*). Thus, poles help coordinate lengths across sister k-fibers such that chromosomes can move within the metaphase spindle while maintaining spindle length.

Together, our findings indicate that spindle poles are not required to globally maintain k-fiber length. Instead, individual k-fibers can locally maintain their length scale over time, and poles and global pole-focusing forces are needed to coordinate k-fiber lengths within the cell and across sister k-fibers, organizing the spindle's structure in space and time.

## Kinetochore-fibers recover their lengths without focused poles

We have shown that k-fibers can establish and maintain their length independently of poles and pole-focusing forces, but cannot properly organize their lengths within the spindle across space and time. While unfocused k-fibers within a cell maintain their average length over time, we sought to determine whether they can recover their length without focused poles, that is, whether they actively adjust and recover their length if shortened below their steady-state length. First, we used laser ablation to acutely cut and shorten k-fibers and then imaged their regrowth compared to unablated k-fibers (*Figure 2A–D*, *Figure 2—video 1*). Mean k-fiber lengths in unfocused spindles before ablation appeared to be shorter (*Figure 2D*); however, this was due to not capturing the full length of k-fibers in a single z-plane while imaging ablated k-fibers. Indeed, length analysis of full z-stacks from unfocused spindles before ablation yielded an indistinguishable mean k-fiber length compared to control k-fibers in *Figure 1E* (*Figure 2—figure supplement 1*). Thus, ablated k-fibers were compared to their unablated neighbors as internal controls. Ablation generates new microtubule minus-ends on the shortened k-fiber stub, which recruit NuMA and dynein to reincorporate them back into the pole in control cells (*Elting et al., 2014*; *Sikirzhytski et al., 2014*). As expected, control ablated k-fibers were transported toward poles and did so while growing back rapidly following ablation, at 0.85 ± 0.09 μm/min on average in the first 5 min (*Figure 2E*). Unfocused k-fibers also grew back, though more slowly at 0.38 ± 0.42 μm/min on average (*Figure 2E*). They took longer to grow back to the mean length of unablated neighbor k-fibers neighbor k-fibers (*Figure 2F*). Thus, focused poles and pole-focusing forces are not required for k-fibers to recover their lengths, but are required for rapid length recovery. The latter is consistent with the idea that force on k-fiber ends favors k-fiber growth (*Dumont and Mitchison, 2009b*; *Long et al., 2020*; *Nicklas and Staehly, 1967*). Ultimately, k-fibers can adapt to length changes and maintain a steady-state length locally, without poles.

To test whether neighboring k-fibers or existing microtubule networks provide information for length maintenance, we treated spindles with nocodazole to depolymerize all microtubules, then washed it out and imaged spindle reassembly (*Figure 2G*, *Figure 2—video 2*). After 10 min, control spindle k-fibers had regrown to within 1 μm of their original length, albeit shorter on average, and unfocused spindle k-fibers fully recovered their average length and grew back into an unfocused state (*Figure 2G–I*, *Figure 2—video 2*). Both control and unfocused spindles could enter anaphase after

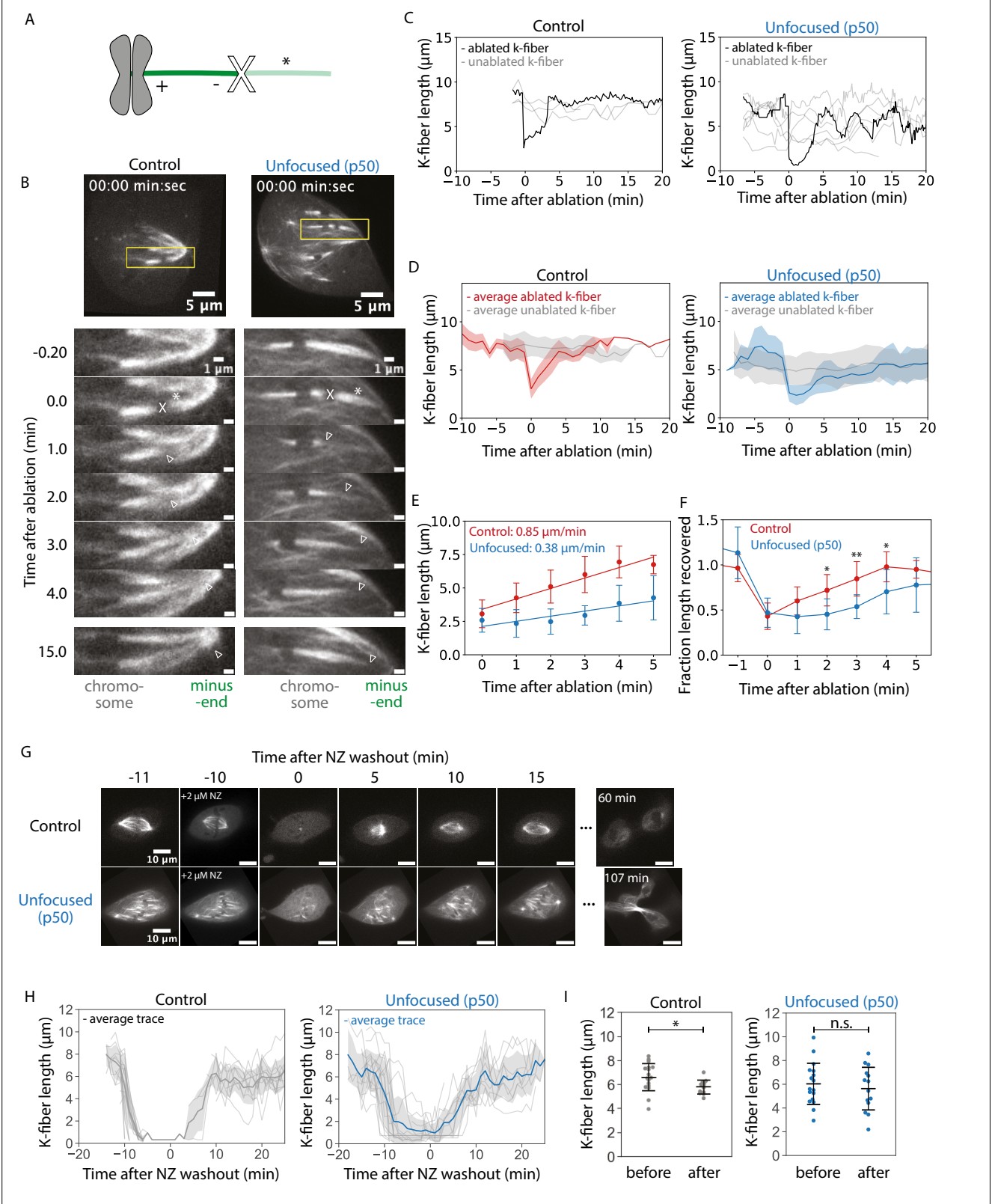

**Figure 2.** Kinetochore-fibers recover their lengths without focused poles. See also *Figure 2—videos 1 and 2*. (**A**) Schematic of a k-fiber after ablation at position X. The k-fiber stub still attached to the chromosome persists with a new minus-end (dark green). The k-fiber segment closer to the pole with a new plus-end depolymerizes away (light green, *). (**B**) Representative confocal timelapse images of PtK2 k-fibers with GFP-α-tubulin and mCherry-p50 (in unfocused only). K-fibers were laser ablated at *t* = 0 (X) and followed over time. Empty arrowheads mark newly created minus-ends. (**C**) K-fiber

*Figure 2 continued*

lengths over time in a representative control and unfocused spindle. Gray traces represent unablated k-fibers. The ablated k-fiber is plotted in black. (**D**) Binned and averaged k-fiber lengths over time for ablated control and unfocused spindles. The average length of non-ablated k-fibers is plotted in gray, the average of ablated k-fibers in red for control and blue for unfocused. Shaded colors indicate ±1 standard deviation for their respective condition (Control: $N$ = 7 cells, $n$ = 8 ablated k-fibers, $m$ = 26 non-ablated k-fibers; Unfocused: $N$ = 6 cells, $n$ = 8 ablated k-fibers, $m$ = 31 non-ablated k-fibers). (**E**) Average growth rates of k-fibers immediately following ablation. Linear regression was performed on binned k-fiber lengths during the first 5 min following ablation (Control: 0.85 ± 0.09 μm/min, Unfocused: 0.38 ± 0.42 μm/min, p = 0.023). (**F**) Fraction of length recovered following ablation relative to the mean of unablated k-fibers in control and unfocused k-fibers. The average trace for unablated k-fibers in D was averaged over time and ablated lengths were normalized to this value. Times with statistically significant differences in length recovery are denoted by *. (**G**) Representative confocal timelapse images of PtK2 spindles with GFP-α-tubulin (in control and unfocused) and mCherry-p50 (in unfocused only), with 2 μM nocodazole added at −10 min and washed out at t = 0. (**H**) Lengths of k-fibers over time during nocodazole washout. All k-fibers are shown with the average trace plotted with ±1 standard deviation shaded in light gray (Control: $N$ = 3 cells, $n$ = 28 k-fibers; Unfocused: $N$ = 4 cells, $n$ = 23 k-fibers). (**I**) Mean k-fiber lengths before nocodazole and after washout in control and unfocused spindles (Control before: 6.58 ± 1.15 μm, $n$ = 17; Control after: 5.76 ± 0.57 μm, $n$ = 12, p = 0.02; Unfocused before: 6.03 ± 1.73 μm, $n$ = 17; Unfocused after: 5.63 ± 1.80 μm, $n$ = 14, p = 0.55). Numbers are mean ± standard deviation. Significance values determined by Welch's two-tailed $t$-test denoted by * for p < 0.05, ** for p < 0.005, and *** for p < 0.0005.

The online version of this article includes the following video, source data, and figure supplement(s) for figure 2:

**Source data 1.** This spreadsheet contains the data used to generate plots 2C, 2D, 2E, 2F, and 2S1.

**Source data 2.** This spreadsheet contains the data used to generate plots 2H and 2I.

**Figure supplement 1.** Kinetochore-fiber lengths before ablation.

**Figure 2—video 1.** Ablating kinetochore-fibers: control vs. unfocused spindle.

https://elifesciences.org/articles/85208/figures#fig2video1

**Figure 2—video 2.** Spindle assembly after nocodazole washout: control vs. unfocused spindle.

https://elifesciences.org/articles/85208/figures#fig2video2

nocodazole washout (*Figure 2G*, *Figure 2—video 2*). Thus, cells lacking pole-focusing forces in metaphase can self-assemble unfocused spindles with k-fibers of about the same length as control k-fibers. This supports a model of k-fibers regulating their own lengths without cues from pre-existing microtubule networks or neighboring k-fibers to build a bi-oriented spindle of the correct length scale.

## Kinetochore-fibers exhibit reduced end dynamics in the absence of poles and pole-focusing forces

Given that k-fibers can maintain (*Figure 1*) and recover (*Figure 2*) their mean length without poles and pole focusing-forces—albeit regrowing more slowly—we asked whether unfocused k-fibers are dynamic and whether they have reduced dynamics. If dynamics are locally set for each individual k-fiber, dynamics should not change without poles or pole-focusing forces; if dynamics are set by global pole-focusing forces, we expect different dynamics without poles. In principle, dynamics can be probed using autocorrelation analysis, which reveals the timescale over which k-fibers 'remember' their length. If k-fibers were less dynamic and their lengths changed more slowly, this would result in stronger autocorrelation and autocorrelation for a longer period. Indeed, this is what we observed in unfocused k-fibers compared to control, consistent with unfocused k-fibers having reduced dynamics (*Figure 3A*). We thus sought to directly measure k-fiber end dynamics and flux.

At metaphase, k-fiber ends are dynamic, with poleward flux associating with net polymerization at plus-ends and apparent depolymerization at minus-ends (*Mitchison, 1989*). Spindle poles have been proposed to regulate minus-end dynamics (*Dumont and Mitchison, 2009b*; *Gaetz and Kapoor, 2004*; *Ganem and Compton, 2004*). To measure k-fiber dynamics, we introduced a bleach mark on a k-fiber and tracked its position over time relative to k-fiber minus-ends (*Figure 3B–D*, *Figure 3—video 1*). In control spindles, the mark approached minus-ends at a rate of 0.55 ± 0.29 μm/min, consistent with previous reports (*Figures 3D and 4D*, *Cameron et al., 2006*; *Mitchison, 1989*). In unfocused spindles, the mark approached minus-ends much slower at a rate of 0.13±0.15 μm/min (*Figures 3D and 4D*). These findings are in contrast to work in *Xenopus* showing that dynein inhibition through p50 overexpression does not impact the flux rate in the central spindle (*Yang et al., 2008*), but are supported by work in *Xenopus* and in mammals showing that dynein contributes to poleward transport (*Burbank et al., 2007*; *Lecland and Lüders, 2014*; *Steblyanko et al., 2020*). Thus, spindle poles or pole-focusing forces are required for fast k-fiber end dynamics, likely contributing to less efficient k-fiber length maintenance in unfocused spindles.

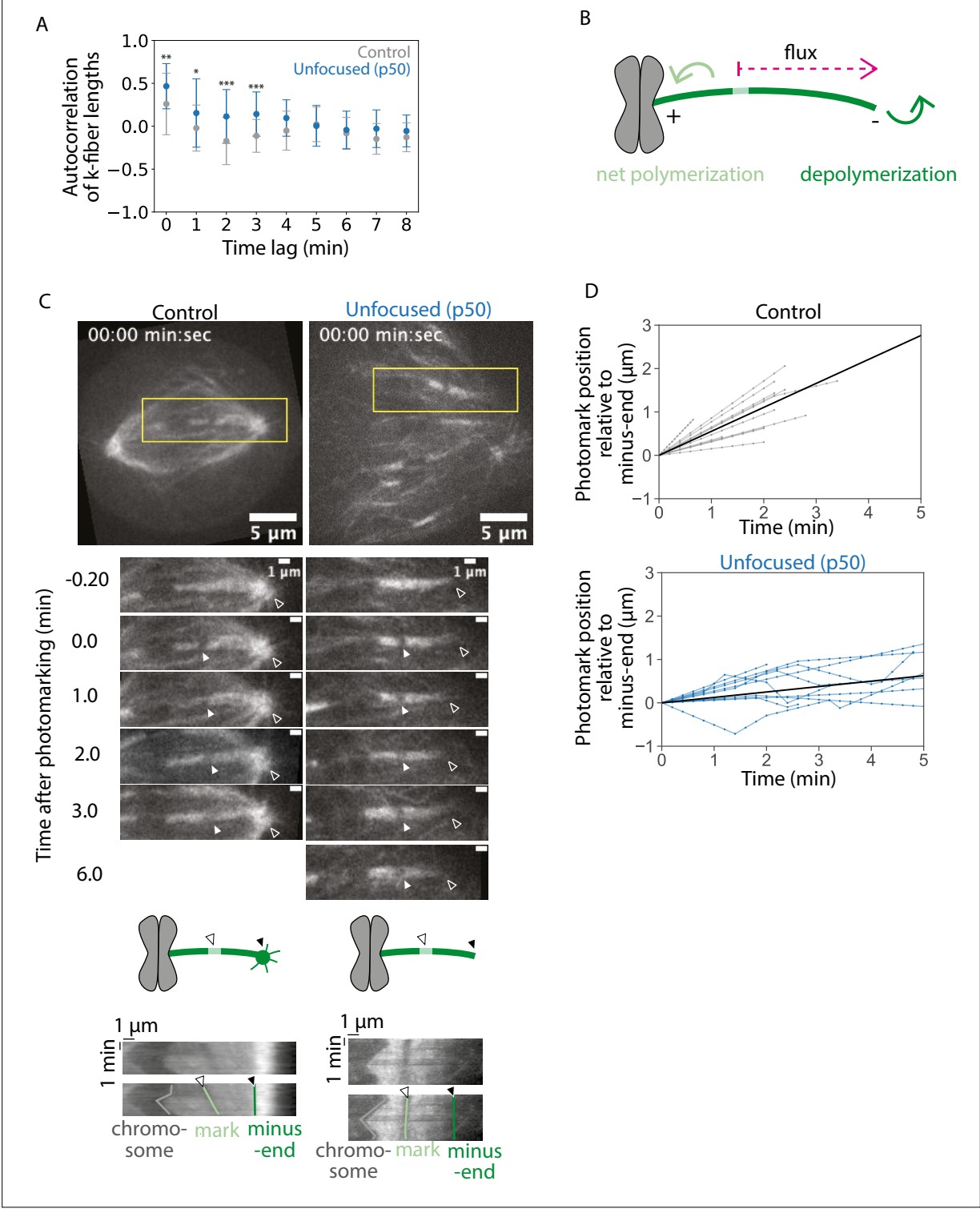

**Figure 3.** Kinetochore-fibers exhibit reduced end dynamics in the absence of poles and pole-focusing forces. See also *Figure 3—video 1*. (**A**) Autocorrelation of k-fiber lengths over time from *Figure 1H* for control and unfocused k-fibers. Calculations and statistical analysis were performed using built-in Mathematica functions, where * indicates $p < 0.05$. (**B**) Schematic of a photomark (light green) on a k-fiber (dark green). The dotted arrow shows the direction the photomark moves with flux in control, where displacement of the mark toward the minus-end increases over time. Net end dynamics are shown by curved arrows (equal at steady state). (**C**) Representative confocal timelapse images of PtK2 k-fibers with GFP-α-tubulin (in control and unfocused) and mCherry-p50 (in unfocused only). A bleach mark was made at time = 0 and followed over time (filled arrowhead). Empty arrowheads indicate minus-ends. Below: Kymographs of the above images. Each row of pixels represents a max-intensity projection of a 5-pixel high

*Figure 3 continued on next page*

Figure 3 continued

stationary box drawn around the k-fiber at one time point (yellow box). (**D**) Minus-end dynamics, where photomark position over time describes how the mark approaches the k-fiber's minus-end over time in control and unfocused k-fibers. Each trace represents one mark on one k-fiber. To measure flux as defined by minus-end depolymerization, the movement of the photomark toward the minus-end was plotted over time. Line with the average slope is drawn in black (Control: $N = 8$ cells, $n = 12$ k-fibers; Unfocused: $N = 8$ cells, $n = 11$ k-fibers). Numbers are mean ± standard deviation. Significance values determined by Welch's two-tailed $t$-test denoted by n.s. for $p \geq 0.05$, * for $p < 0.05$, ** for $p < 0.005$, and *** for $p < 0.0005$.

The online version of this article includes the following video and source data for figure 3:

**Source data 1.** This spreadsheet contains the data used to generate plot 3D.

**Figure 3—video 1.** Photobleaching kinetochore-fibers to measure microtubule flux: control vs. unfocused spindle.

https://elifesciences.org/articles/85208/figures#fig3video1

## Kinetochore-fibers tune their end dynamics to recover length, without pole-focusing forces

The fact that unfocused k-fibers grow back to a steady-state length after being acutely shortened (*Figure 2*) suggests that they can tune their dynamics after shortening. We thus sought to determine the physical mechanism for length recovery (*Figure 4A*). One model is that minus-end depolymerization stops or slows—for example, pole-based depolymerization dynamics are lost while k-fiber minus-ends appear separated from the pole (*Dumont and Mitchison, 2009b*; *Long et al., 2020*). Another model is that plus-end polymerization increases, which could occur in either a force-dependent manner (*Akiyoshi et al., 2010*; *Dumont and Mitchison, 2009b*; *Long et al., 2020*; *Nicklas and Staehly, 1967*) or a length-dependent manner (*Dudka et al., 2019*; *Mayr et al., 2007*; *Stumpff et al., 2008*; *Varga et al., 2006*). Notably, we find that k-fibers can grow back after ablation (*Figure 2E*) at a rate faster than poleward flux and associated minus-end dynamics in both control and unfocused spindles (0.85 ± 0.09 vs. 0.55 ± 0.29 µm/min in control, 0.38 ± 0.42 vs. 0.13 ± 0.15 µm/min in unfocused) (*Figures 2E and 4D*). Thus, even if minus-end dynamics were suppressed, this would not be sufficient to account for the k-fiber regrowth we observe after ablation, with or without pole-focusing forces.

To directly test how changes in k-fiber length regulate end dynamics, and if this mechanism depends on pole-focusing forces, we ablated a k-fiber and introduced a photobleach mark on it in control and unfocused spindles (*Figure 4A, B*). In control spindles, the photomark did not detectably approach the minus-end of the k-fiber during its regrowth (*Figure 4B, C*), indicating that suppression of minus-end dynamics contributes to k-fiber regrowth, as in *Drosophila* cells (*Maiato et al., 2004*; *Matos et al., 2009*). Consistent with k-fiber minus-end dynamics being transiently suppressed during regrowth, rather than frozen due to ablation damage, k-fiber minus-ends resumed depolymerization in control spindles after ablation and length recovery (*Figure 4—figure supplement 1*, *Figure 4—video 2*). However, while *Drosophila* k-fibers regrow at the rate of poleward flux, these control mammalian k-fibers regrow faster than the rate of flux, indicating that mammalian k-fibers must additionally increase their plus-end dynamics when shortened to reestablish their steady-state length. In unfocused spindles, the photomark also did not detectably approach the minus-end of the k-fiber during its regrowth (*Figure 4C*), consistent with suppression of any minus-end dynamics, though it was not significantly different from the already slow dynamics and insufficient to account for growth (*Figure 4D*). Thus, k-fibers can tune their plus-end dynamics to recover their length in the absence of dynein-based pole-focusing forces. This supports a model where k-fiber length is not simply regulated by global pole-focusing forces, but by local length-based mechanisms.

## Spindle poles coordinate chromosome segregation and cytokinesis

So far, we have shown that while a focused pole is not required for setting or maintaining k-fiber lengths (*Figures 1 and 2*), it is required for global spindle coordination (*Figure 1*) and robust k-fiber dynamics (*Figures 3 and 4*). To test the functional output of focused spindle poles in mammalian cells, we treated control and unfocused spindles with reversine, an MPS1 inhibitor that forces mitotic cells to enter anaphase, even in the absence of dynein activity required for spindle assembly checkpoint satisfaction (*Santaguida et al., 2010*). Control and unfocused spindles were imaged through anaphase after reversine addition using a single z-plane (*Figure 5A*, *Figure 5—video 1*) and also imaged with z-stacks encompassing the whole spindle once before adding reversine, and 20 min after anaphase onset (*Figure 5B*). In spindles without focused poles, chromatids separated—albeit

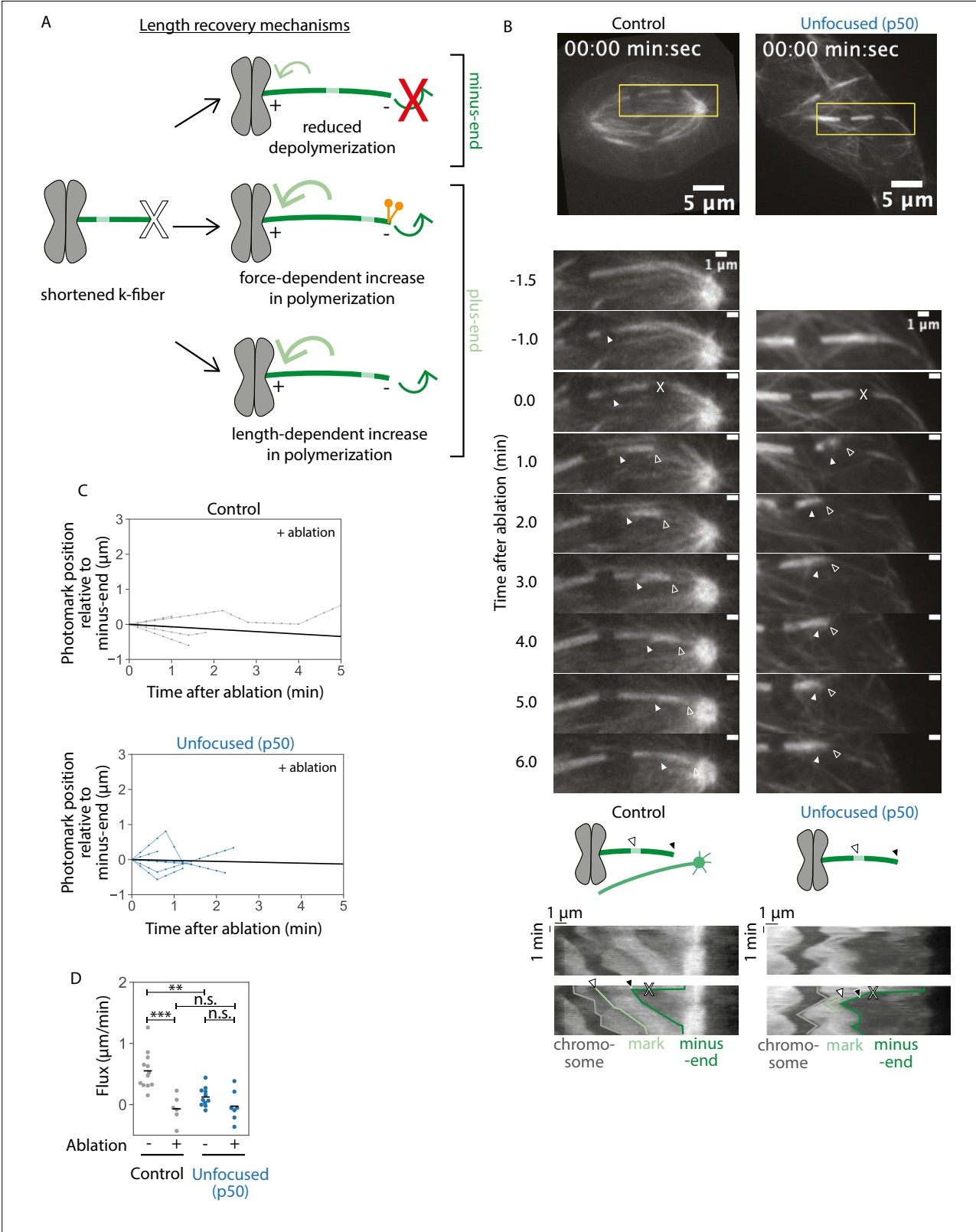

**Figure 4.** Kinetochore-fibers tune their end dynamics to recover length, without pole-focusing forces. See also *Figure 4—video 1*. (**A**) Models describing k-fiber length recovery mechanisms. K-fibers shortened by ablation (X) with a photomark (light green) can potentially grow back in different ways: suppression of minus-end depolymerization (top), increased plus-end polymerization induced by forces such as dynein (middle), or increased polymerization in a length-dependent manner (bottom). (**B**) Representative confocal timelapse images of PtK2 k-fibers with GFP-α-tubulin (in control

*Figure 4 continued on next page*

*Figure 4 continued*

and unfocused) and mCherry-p50 (in unfocused only). Filled arrowhead follows a bleach mark. At $t = 0$, k-fibers were cut with a pulsed laser at higher power (X). Empty arrowhead follows the new k-fiber minus-end. Below: Kymographs of the above images as prepared in *Figure 3C*. (C) Minus-end dynamics were probed by tracking movement of the mark toward the k-fiber's minus-end over time in control and unfocused k-fibers after ablation at $t = 0$. Line with the average slope is drawn in black (Control: $N = 5$ cells, $n = 6$ k-fibers; Unfocused: $N = 7$ cells, $n = 7$ k-fibers). (D) Minus-end dynamics of k-fibers. Flux as measured by rate of photomark movement toward the minus-end with or without ablation in control and unfocused k-fibers. Each point represents the slope of one trace in *Figure 3D* or (C) measured by linear regression (Control: mean flux = 0.55 ± 0.29 μm/min, mean flux after ablation = −0.07 ± 0.20 μm/min; Unfocused: mean flux = 0.13 ± 0.15 μm/min, mean flux after ablation = −0.03 ± 0.23 μm/min; p non-ablated control vs. ablated control = 2.7e−4, p non-ablated control vs. non-ablated unfocused = 5.3e−4, p non-ablated unfocused vs. ablated unfocused = 0.19, p ablated control vs. ablated unfocused = 0.75). Numbers are mean ± standard deviation. Significance values determined by Welch's two-tailed *t*-test denoted by n.s. for $p \geq 0.05$, * for $p < 0.05$, ** for $p < 0.005$, and *** for $p < 0.0005$.

The online version of this article includes the following video, source data, and figure supplement(s) for figure 4:

**Source data 1.** This spreadsheet contains the data used to generate plots 4C and 4D.

**Figure supplement 1.** Minus-end depolymerization resumes after length recovery following ablation.

**Figure supplement 1—source data 1.** This spreadsheet contains the data used to generate plot 4S1.

**Figure 4—video 1.** Ablating and photomarking kinetochore-fibers: control vs. unfocused spindle.
https://elifesciences.org/articles/85208/figures#fig4video1

**Figure 4—video 2.** Photobleaching control kinetochore-fibers after ablation and length recovery.
https://elifesciences.org/articles/85208/figures#fig4video2

at twofold reduced velocities compared to control—in the separating chromatid pairs that could be identified (*Figure 5C*). In the absence of poles or dynein activity, such chromatid separation likely comes from pushing from the spindle center rather than from pulling from the cell cortex (*Vukušić et al., 2017*; *Yu et al., 2019*).

However, major segregation and cytokinetic defects were observed in these cells compared to control, consistent with segregation defects observed in k-fibers disconnected from poles (*van Toorn et al., 2022*; *Sivaram et al., 2009*). Cytokinetic defects and the presence of multiple cytokinetic furrows frequently resulted in the formation of more than two daughter cells in unfocused spindles (*Figure 5D*). Furthermore, chromosome masses were scattered and unequally distributed in these cells, whereby control daughter cells inherited approximately half of the chromosome mass as measured by DNA intensity, but not daughter cells of unfocused spindles (*Figure 5E*). Given that focused mammalian spindles lacking dynein pole-focusing forces and lacking Eg5 proceed through anaphase with much milder defects than we observe here (*Neahring et al., 2021*), we conclude that poles, rather than dynein-based pole-focusing forces, are primarily responsible for these defects. Thus, while many species lack spindle poles, and while unfocused mammalian spindles can still maintain k-fiber length and separate chromatids, spindle poles are essential to coordinate chromosome segregation and cytokinesis in mammalian cells.

## Discussion

Here, we show that in the mammalian spindle, individual k-fibers set and maintain their lengths locally but require the global cue of a focused pole to coordinate their lengths across space and time (*Figure 6*). Our work reveals that pole-less spindles can set and maintain k-fibers at the same mean length as in control, recovering their steady-state lengths if acutely shortened, but they have impaired dynamics and coordination and are unable to properly segregate chromosomes. We propose a model whereby length is an emergent property of individual k-fibers in the spindle, and where spindle poles ensure that this network of k-fibers is highly dynamic and coordinated across space and time to ultimately cluster chromatids into two future daughter cells.

While this work provides insight into k-fiber length establishment and maintenance, what local mechanisms set the k-fiber's length scale remains an open question. We discuss three models. First, concentration gradients centered on chromosomes (*Kalab and Heald, 2008*; *Wang et al., 2011*) could in principle set a distance-dependent activity threshold for spindle proteins that regulate k-fiber dynamics and length. However, it is unclear whether such a gradient with correct length scale and function exists in mammalian spindles. Also, while the globally disorganized structure of unfocused spindles (*Figure 1B, D*) could lead to modified gradients, the mean length of k-fibers is unchanged

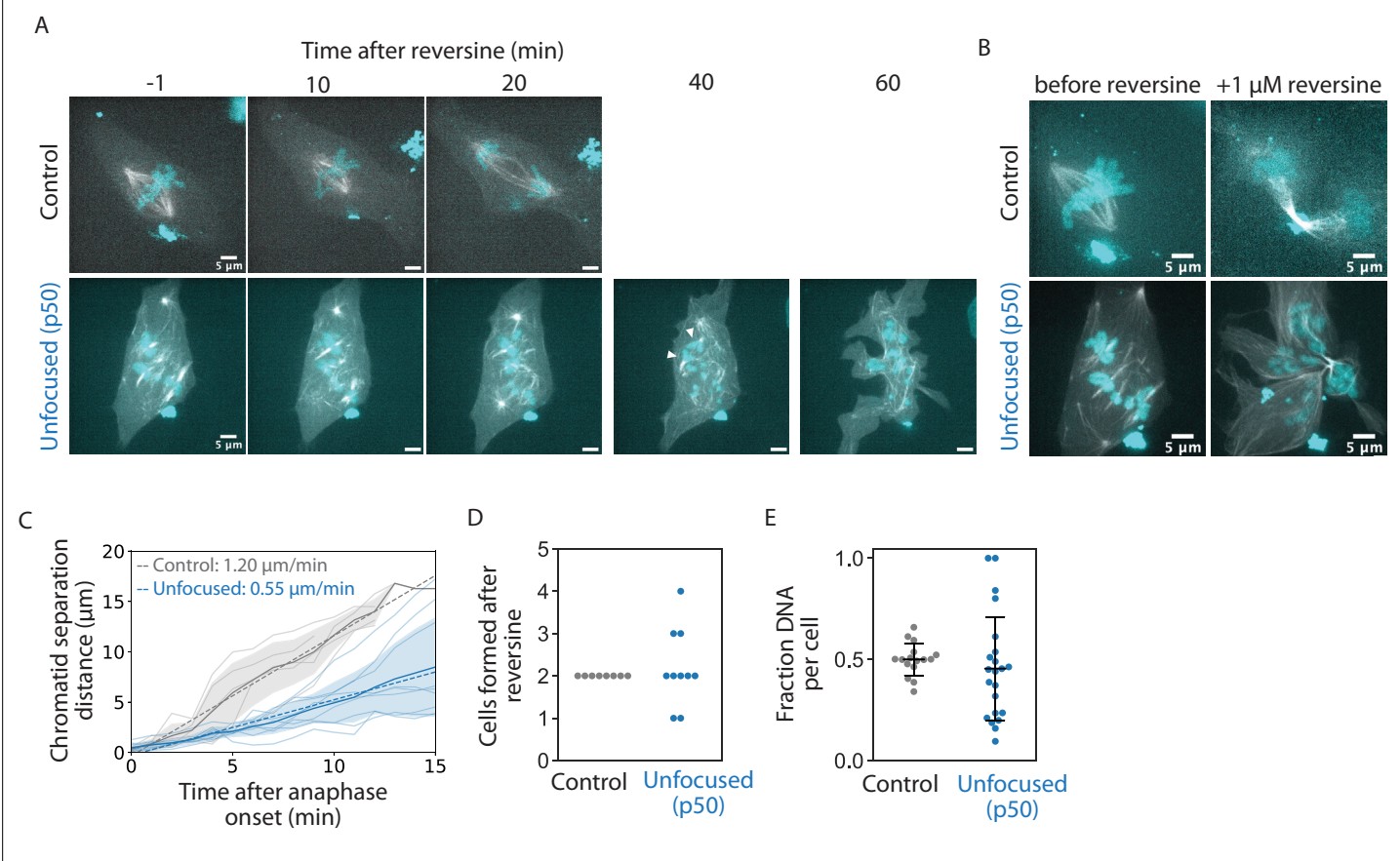

**Figure 5.** Spindle poles coordinate chromosome segregation and cytokinesis. See also *Figure 4—video 1*. (**A**) Representative confocal timelapse images of PtK2 spindles with GFP-α-tubulin (in control and unfocused) and mCherry-p50 (in unfocused only) treated with 0.1 or 0.5 µM SiR-DNA with 1 µM reversine added at *t* = 0. Arrowheads depict an example of sister chromatids separating, later measured in C. (**B**) Max-intensity z-projections before adding reversine and 20 min after anaphase onset for the control and unfocused spindle in A. Figures C–E are from the same dataset (Control: *N* = 8 dividing cells; Unfocused: *N* = 10 dividing cells). (**C**) Sister chromatid separation velocity. For the chromatid pairs that were observed to separate, sister chromatid distance over time was measured for focused and unfocused spindles starting at anaphase onset. Control is plotted in gray, unfocused in blue. Light-colored traces represent one separating chromatid pair, with their average plotted as a dark line with shading representing ±1 standard deviation. The line of best fit for each condition averaged is shown as a dotted line, with their slopes shown (Control: *N* = 4 dividing cells, *n* = 5 chromosome pairs, separation velocity = 1.20 µm/min; Unfocused: *N* = 3 dividing cells, *n* = 9 chromatid pairs, separation velocity = 0.55 µm/min). (**D**) Number of 'cells' formed after cytokinesis in reversine-treated control and unfocused spindles (Control: 2 ± 0 cells; Unfocused: 2.20 ± 0.87 'cells'). (**E**) Fraction of chromosome mass per 'cell' after reversine treatment. Summed z-projections of chromosome masses were used to calculate the fraction of chromosome mass per cell (Control: 0.50 ± 0.08 a.u.; Unfocused: 0.45 ± 0.26 a.u.). Numbers are mean ± standard deviation.

The online version of this article includes the following video and source data for figure 5:

**Source data 1.** This spreadsheet contains the data used to generate plots 5C, 5D, and 5E.

**Figure 5—video 1.** A reversine-treated control spindle undergoing anaphase: control vs. unfocused spindle.

https://elifesciences.org/articles/85208/figures#fig5video1

and length does not correlate with spatial position along both spindle axes (*Figure 1E*, *Figure 1— figure supplement 6*). Second, a lifetime model (*Burbank et al., 2007*; *Conway et al., 2022*) stipulates that length is proportional to microtubule lifetime and the velocity of poleward transport, and is sufficient to predict spindle length in spindles with a tiled array of short microtubules. While the length distribution of individual microtubules in unfocused k-fibers is unknown, this model would predict an exponential distribution of microtubule lengths within a k-fiber (*Brugués et al., 2012*), inconsistent with electron microscopy in control PtK cells (*McDonald et al., 1992*). Moreover, we observed a more than fourfold reduced (and near zero) flux velocity in unfocused spindles (*Figure 3D*), which only a dramatic increase in lifetime could compensate for in this lifetime model. Finally, an 'antenna' model (*Varga et al., 2006*) stipulates that longer k-fibers recruit more microtubule dynamics regulators since

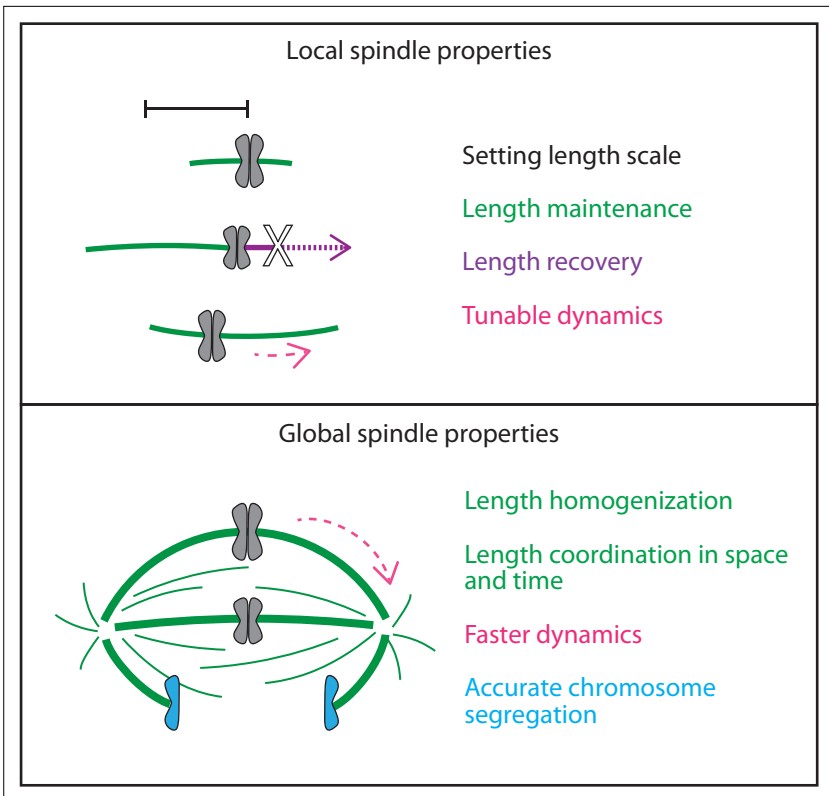

**Figure 6.** Spindle length is a local spindle property and length coordination is a global spindle property. Cartoon summary of spindle properties set locally vs. globally. Setting, maintaining, and recovering length is regulated by individual k-fibers locally, independently of poles and pole-focusing forces. In turn, coordinating lengths across space and time requires global cues from focused poles. In sum, spindle length emerges locally, but spindle coordination emerges globally.

they have a longer microtubule antenna to land on. For example, in mammalian spindles, the microtubule depolymerase Kif18A binds k-fibers in a length dependent way and exhibits length-dependent depolymerase activity, being more active on long k-fibers and thereby shortening them (*Mayr et al., 2007*; *Stumpff et al., 2008*). Given that this local antenna model is consistent with our current observations, testing in unfocused spindles whether k-fiber growth rate indeed changes with k-fiber length and testing the role of dynamics regulators in length establishment and maintenance represent important future directions.

Our findings suggest that in response to length changes, k-fibers regulate their plus-end dynamics in an analog manner and their minus-end dynamics in a digital manner. In unfocused spindles, we have shown that the regrowth of shortened k-fibers is driven by an increase in plus-end polymerization, and that this occurs in response to length changes, not simply dynein-based force changes (*Figure 4*). Consistently, longer k-fibers grow more slowly than shorter ones in a titratable manner in human spindles (*Conway et al., 2022*). The regulation mechanisms above are all analog in nature. In turn, after ablation, we always observed a near-absence of minus-end dynamics. This reduction in flux was only large enough in control k-fibers to observe statistical significance, though unfocused k-fibers appeared to follow the same trend (*Figure 4C and D*). This is consistent with a switchlike mechanism turning depolymerization on or off, proposed on the basis that tension on k-fibers turns off apparent minus-end depolymerization (*Dumont and Mitchison, 2009b*; *Long et al., 2020*). The mechanism behind such digital regulation is not known. One possibility is that a proximal pole structure is required to recruit active microtubule depolymerases, such as Kif2a (*Gaetz and Kapoor, 2004*; *Ganem et al., 2005*), to k-fiber minus-ends. In unfocused spindles without a pole, k-fibers would be less dynamic (*Figure 3D*) based on having fewer depolymerases at their minus-ends. In physical perturbation experiments where k-fibers are separated from the pole center, their apparent minus-end depolymerization would stop (*Dumont and Mitchison, 2009b*; *Long et al., 2020*) based on a too-distant depolymerase

pool and thus fewer depolymerases at minus-ends. Interestingly, Kif2a can drive spindle scaling in *Xenopus* meiotic spindles (*Wilbur and Heald, 2013*).

In principle, the concomitant loss of dynein-mediated pole-focusing forces and spindle poles makes it difficult to disentangle the role of each in regulating spindle coordination, maintenance, and function in our findings. However, recent work has revealed that mammalian spindles can achieve similar architecture whether or not dynein (or its recruiter NuMA) is knocked out (*Neahring et al., 2021*). This suggests that the severe defects in spindle coordination (*Figures 1 and 5*) and maintenance (*Figure 2*) observed in p50-unfocused spindles are more likely due to the loss of spindle poles than due to the loss of dynein activity per se. Though we cannot exclude it, this also suggests that the findings we make in unfocused spindles are not due changes in activity of the dynein population at kinetochores. Additionally, centrosomes are disconnected from the spindle (*Figure 1—videos 2 and 3*), ruling out contributions from centrosomes (*Khodjakov et al., 2000*) or astral microtubules on k-fiber length regulation at metaphase. Mammalian spindle poles are also required for spindle positioning (*Kiyomitsu and Cheeseman, 2012*) and have been proposed to help segregate centrosomes (*Friedländer and Wahrman, 1970*). More work is needed to understand the evolution and function of spindle poles across species and, more broadly, the diversity of spindle architectures across evolution.

We propose that this biological blueprint, where k-fibers locally set and maintain their individual length and poles coordinate them globally, robustly builds a complex yet dynamic spindle. For example, we have shown that while k-fibers establish their mean lengths locally, global cues homogenize them (*Figure 1E, G*). We put forward the idea that the structural integrity and flexible remodeling of other higher-order structures may also rely on individual parts having all the necessary intrinsic information and self-organization to get the correct linear architecture, with global cues organizing these parts in space and time. More broadly, our work highlights how self-organization at local scales and coordination at global scales can work together to build emergent complex biological structures.

Videos are displayed with optimal brightness and contrast for viewing.

# Materials and methods

**Key resources table**

| Reagent type (species) or resource | Designation | Source or reference | Identifiers | Additional information |
|---|---|---|---|---|
| Cell line (*P. tridactylus*, male) | PtK2 | Gift from T. Mitchison, Harvard University | PMID:1633624 | Kidney epithelial |
| Cell line (*P. tridactylus*, male) | HaloTag-tubulin PtK2 | This paper | | Kidney epithelial |
| Cell line (*H. sapiens*, female) | RPE1 | ATCC | ATCC Cat#CRL-4000; RRID: CVCL_4388 | Retina, epithelial |
| Chemical compound, drug | Nocodazole | Sigma | M1404 | Final concentration 2 µM |
| Chemical compound, drug | Reversine | Sigma | R3904 | Final concentration 1 µM |
| Chemical compound, drug | Viafect | ProMega | E4981 | 1:6 ratio of Viafect:DNA used |
| Chemical compound, drug | Janelia Fluor 646 | Janelia | 6148 | Final concentration 100 nM |
| Chemical compound, drug | SiR-DNA | Spirochrome | SC007 | Final concentration 0.1–0.5 µM with 1 µM verapamil |
| Chemical compound, drug | SiR-tubulin | Spirochrome | SC002 | Final concentration 0.1 µM with 1 µM verapamil |
| Recombinant DNA reagent | pLV-β-tubulin-HaloTag (plasmid) | This paper | | Lentiviral plasmid. Progenitors: Addgene #114021 (Geert Kops) and Addgene #64691 (Yasushi Okada) |

*Continued on next page*

*Continued*

| Reagent type (species) or resource | Designation | Source or reference | Identifiers | Additional information |
|---|---|---|---|---|
| Recombinant DNA reagent | pLV-mCherry-p50 (plasmid) | This paper | | Lentiviral plasmid. Progenitors: Addgene #114021 (Geert Kops) and mCherry-p50 (PMID:19196984) |
| Recombinant DNA reagent | eGFP-α-tubulin (plasmid) | Michael Davidson collection given to UCSF; *Rizzo et al., 2009* | Addgene Plasmid #56450 | |
| Recombinant DNA reagent | mCherry-p50 (plasmid) | Gift from M. Meffert, Johns Hopkins University; | PMID:19196984 | |
| Recombinant DNA reagent | β-Tubulin HaloTag (plasmid) | Addgene; *Uno et al., 2014* | Addgene Plasmid #64691 | |
| Software, algorithm | FIJI | FIJI; *Schindelin et al., 2012* | ImageJ version 2.1.0 | |
| Software, algorithm | Wolfram Mathematica | Wolfram Mathematica | Version 13.0 | |
| Software, algorithm | MetaMorph | MDS Analytical Technologies | Version 7.8 | |
| Software, algorithm | Micro-Manager | Micro-Manager; *Edelstein et al., 2010* | Version 2.0.0 | |
| Software, algorithm | Python | Python | Version 3.8.1 | Spyder IDE version 4.1.5 |

## Cell culture

All work herein was performed using wild-type PtK2 cells (*P. tridactylus*, male, PMID:1633624, kidney epithelial, gift from Tim Mitchison, Harvard University) unless otherwise stated. PtK2 cells were cultured in MEM (11095; Thermo Fisher, Waltham, MA) supplemented with sodium pyruvate (11360; Thermo Fisher), non-essential amino acids (11140; Thermo Fisher), penicillin/streptomycin, and 10% heat-inactivated fetal bovine serum (10438; Thermo Fisher). The cell line tested negative for mycoplasma, and while we did not authenticate it, its cell behavior and growth characteristics were similar to those reported for the parental PtK2 cell line, whose transcriptome we sequenced (*Udy et al., 2015*). Cells were maintained at 37°C and 5% $CO_2$. hTERT-RPE1 cells (*H. sapiens*, female, ATCC Cat#CRL-4000, RRID: CVCL_4388, retinal epithelial) were cultured in Dulbecco's Modified Eagle Medium/Nutrient Mixture F-12 with GlutaMAX (11320; Thermo Fisher) supplemented with penicillin/streptomycin and 10% fetal bovine serum. This cell line was not authenticated by short tandem repeat (STR) profiling but tested negative for mycoplasma.

To visualize microtubules, PtK2 cells were transfected with eGFP-α-tubulin (Clontech) using Viafect (Promega) unless otherwise noted. To inhibit dynein, PtK2 or RPE1 cells were additionally transfected or lentivirally infected with mCherry-p50 (a gift from Mollie Meffert, Johns Hopkins University; *Shrum et al., 2009*). Transient transfections were prepared in a 100-μl reaction mix per 35 mm dish, including a 1:6 ratio of DNA to Viafect, OptiMEM media up to 100 μl, and eGFP-α-tubulin (0.7 μg) or both eGFP-α-tubulin (0.4 μg) and mCherry-p50 (0.5 μg), and added 3–4 days prior to imaging.

## Lentiviral plasmids and cell line construction

The coding sequences of β-tubulin-HaloTag (Addgene #64691) and mCherry-p50 were cloned into a puromycin-resistant lentiviral vector (Addgene #114021) using Gibson assembly. Lentivirus for each construct was produced in HEK293T cells. To generate the stable polyclonal β-tubulin-HaloTag PtK2 cell line (*Figure 1A*), wild-type PtK2 cells were infected with β-tubulin-HaloTag virus and selected using 5 μg/ml puromycin. Because p50 overexpression disrupts cell division, mCherry-p50 lentivirus was used to transiently infect each 35 mm dish 3–4 days prior to imaging (*Figure 1A*).

## Imaging

PtK2 or RPE1 cells were plated on 35 mm #1.5 coverslip glass-bottom dishes coated with poly-D-lysine (MatTek, Ashland, MA) and imaged. The cells were maintained at 30–37°C in a stage top incubator (Tokai Hit, Fujinomiya-shi, Japan). Two similar inverted spinning-disk confocal (CSU-X1; Yokogawa Electric Corporation) microscopes (Eclipse TI-E; Nikon) with the following components were used for live-cell imaging: head dichroic Semrock Di01-T405/488/561/647, head dichroic Semrock

Di01-T405/488/561, ×100 1.45 Ph3 oil objective, a ×60 1.4 Ph3 oil objective, 488 nm (100, 120, or 150 mW), 561 nm (100 or 150 mW), and 642 (100 mW) nm diode lasers, emission filters ET525/36M (Chroma Technology) for GFP fluorophore imaging, ET630/75 M for mCherry, and ET690/50M for JF 646 (Chroma Technology), a perfect focus system (Nikon, Tokyo, Japan), an iXon3 camera (Andor Technology, 105 nm/pixel using ×100 objective at bin = 1), and a Zyla 4.2 sCMOS camera (Andor Technology, 65.7 nm/pixel using ×100 objective at bin = 1). For imaging, 400 ms exposures were used for phase contrast and 50–100 ms exposures were used for fluorescence. Cells were imaged at 30°C (by default) or 37°C to speed up slower processes (*Figures 1A, 2G, H, and 5*), 5% $CO_2$ in a closed, humidity-controlled Tokai Hit PLAM chamber. Cells were imaged via MetaMorph (7.8, MDS Analytical Technologies) or Micro-Manager (2.0.0).

Spindle assembly videos (*Figure 1A*, *Figure 1—videos 1 and 2*) were captured using a ×60 objective for a wider field of view, selecting approximately 20 stage positions and imaging overnight at 37°C for 8–10 hr. To capture unfocused spindle assembly, positions containing cells expressing moderate-to-high levels of mCherry-p50 relative to other cells on the dish were selected. Spindles over time were imaged with 1 µm z-slices every minute to avoid photodamage (*Figure 1A, H*). Volumetric spindle images were taken using a ×100 objective, with z-slices 0.3 µm apart encompassing the whole spindle (*Figures 1B and 5B*, *Figure 5B*, *Figure 2—figure supplement 1*). To visualize DNA, 0.1–0.5 µM SiR-DNA (Spirochrome) with 1 µM verapamil were added at least 30 min prior to imaging (*Figure 5*). To visualize microtubules, 100 nM JF 646 was added to HaloTag-tub PtK2 cells at least 30 min prior to imaging (*Figure 1A*).

## Photobleaching and laser ablation (Figures 2–4)

Photobleaching and laser ablations were performed using 514 or 551 nm ns-pulsed laser light and a galvo-controlled MicroPoint Laser System (Andor, Oxford Instruments) operated through Meta-Morph or Micro-Manager. Single z-planes were chosen to pick the clearest k-fiber visible from plus- to minus-end, parallel to the coverslip, that was long enough to ablate. Non-ablated unfocused k-fibers in the same imaging plane were not necessarily parallel to the coverslip, so their full length was not always captured in the single z-plane due to tilt. Photobleaching was performed by firing the laser at the lowest possible power to make a visible bleach mark (~20% of total power), whereas ablations were performed at the lowest possible power to fully cut a k-fiber (~60% of total power). K-fiber ablations were verified by observing complete depolymerization of newly created plus-ends, relaxation of interkinetochore distance, or poleward transport of k-fiber stubs (control only). When firing the laser, 1–3 areas around the region of interest were targeted and hit with 5–20 pulses each. Ablations were imaged using one z-plane every 12 s to assay short-term dynamics, then switching to every 1 min after approximately 10 min following ablation to avoid phototoxicity.

## Nocodazole washout (Figure 2)

Z-planes containing the highest number of clearly distinguishable k-fibers, that were parallel to the coverslip, were chosen for imaging. 2 µM nocodazole was swapped into dishes using a transfer pipet while imaging. After 10 min to depolymerize microtubules, dishes were washed 10× in prewarmed media to remove nocodazole and allow spindle reassembly. Spindles were imaged at one z-plane every min to avoid phototoxicity during spindle recovery. To measure k-fiber lengths before nocodazole addition, individual k-fiber traces were averaged over time before drug addition (≤−10 min). K-fiber lengths after drug washout were averaged over time after spindles reached a steady-state length (≥10 min), subtracting centrosome radius for control k-fibers during these times.

## Reversine treatment (Figure 5)

Metaphase spindles were volumetrically imaged with a z-step of 0.3 µm across whole live spindles before reversine addition. The media was then swapped to similar media containing 1 µM reversine and imaged at a single z-plane. 20 min after anaphase onset, cells were again imaged volumetrically as previously described.

## Image analysis

Feature tracking, spindle architecture measurements, and statistical analyses were done in FIJI and Python unless otherwise stated. Videos and images are displayed with optimal brightness and contrast for viewing.

## Spindle major and minor axes length (Figure 1D, Figure 1—figure supplement 4A)

Spindle minor and major axes lengths were determined by cropping, rotating, then thresholding spindle images with the Otsu filter using SciKit. Ellipses were fitted to thresholded spindles to approximate the length of their major and minor axes using SciKit's region properties measurement (*Figure 1—figure supplement 4A*). In control spindles, the major axis corresponded to spindle length along the pole-to-pole axis, and the minor axis corresponded to spindle width along the metaphase plate axis. However, unfocused spindles were disorganized along both axes to the extent where the minor axis did not always correspond to the metaphase plate axis. Thus, *Figure 1D* reports 'spindle minor axis length' and 'spindle major axis length' rather than 'spindle width' and 'spindle length'. Furthermore, it is worth noting that in unfocused spindles, spindle length is decoupled from k-fiber length because of k-fiber disorganization along both axes. Thus, spindle length was not measured in unfocused spindles, but individual k-fiber length was measured as described below.

## K-fiber length (Figures 1 and 2, Figure 1—figure supplement 4B, C)

For k-fiber length measurements at a single time point, z-stacks of live spindles were taken with a step size of 0.3 µm across the entire spindle (*Figure 1B*). Individual k-fibers were measured using a maximum intensity z-projection of only the slices where that k-fiber was in focus (*Figure 1—figure supplement 4B*). Line profiles were then measured by drawing ROIs in FIJI with a spline fit line of width 15 pixels, spanning from plus-ends at the start of tubulin intensity next to the chromosome toward minus-ends, using the minimum number of points to recapitulate the curve of the k-fiber (*Figure 1—figure supplement 4B*). The 3D length was then estimated with the Pythagorean theorem, using the length of the k-fiber's ROI and the *z*-height of the slices it spanned (*Figure 1E–G, Figure 1—figure supplement 4C*). For control k-fibers, the end of the ROI spanning the k-fiber was defined as the center of the pole, and centrosome radius was subtracted to estimate true k-fiber length (*Figures 1E–G, I–N and 2C–F, H, I*). Since minus-ends of focused k-fibers are not distinguishable in a pole and typically terminate within 2 µm of centrosomes (*McDonald et al., 1992*), centrosome radius was approximated by drawing line scans through focused poles and measuring the half width at half max intensity (*Figure 1—figure supplement 5*). This approximation was used for all subsequent length measurements. For unfocused and ablated k-fibers, minus-ends were defined as the farthest point of visible tubulin intensity corresponding to that k-fiber. Lengths of ROIs were calculated and plotted in Python. K-fiber lengths over time were measured as described above, but from videos with single imaging planes or from max-intensity projections based on a step size of 1 µm across the volume of the spindle. K-fiber lengths were then measured using ROIs of width 5 pixels for k-fibers whose plus- and minus-ends were visible across at least 5 frames (k-fiber lengths over time, *Figure 1H–N* and ablated k-fibers, *Figure 2C–F*). K-fiber lengths were binned by minute for aggregate analyses.

In k-fibers following ablation, centrosome size was subtracted only when control k-fibers were reincorporated into the pole and the ablated minus-end was no longer visible. To calculate growth rates for k-fiber lengths over time, linear regression was performed using SciPy on binned k-fiber lengths for those with data at time points 0–6 min. One control k-fiber was excluded from growth rate analysis based on these criteria.

## Spatial correlation analysis (Figure 1—figure supplement 6)

K-fiber positions in spindles were quantified latitudinally and longitudinally, then correlated to length. To approximate the metaphase plate axis, a line of best fit was drawn through kinetochore positions in a cell as approximated by the positions of k-fiber plus-ends. Only the positions of k-fiber plus-ends whose sister k-fibers were also measured were used to calculate spindle axes. In control spindles, the metaphase plate axis corresponded to the spindle width axis. The long spindle axis was determined by drawing a line perpendicular to the metaphase plate axis through the average kinetochore position. In control spindles, this long spindle axis corresponded to the pole-to-pole axis. Distance from

each k-fiber's plus-end to the long axis was measured, and then binned into 'inner' k-fibers if ≤2 μm and 'outer' if ≥3 μm.

Alignment scores were calculated based on the distance from each k-fiber's plus-end to the metaphase plate axis, and then given either a negative or positive sign depending on whether the k-fiber was 'over-aligned' (see yellow control example in *Figure 1—figure supplement 6B*) with longer expected lengths or 'under-aligned' (see blue control example in *Figure 1—figure supplement 6B*). K-fibers were categorized as either over- or under-aligned based on the relative positions of their plus- and minus-ends. If both ends were on the same side of the metaphase plate, that is the k-fiber was under-aligned and fully on one side of the metaphase plate, the distance from the plus-end to the metaphase plate was recorded as positive. If the plus- and minus-ends were on opposite sides of the metaphase plate axis, that is the k-fiber was over-aligned and crossing the metaphase plate axis, the distance from the plus-end to the metaphase plate was recorded as negative. A perfectly aligned pair of kinetochores would each have an alignment score of approximately +1. This method of assigning alignment scores was sufficient to accurately categorize all control k-fibers as over- or under-aligned. However, there were rare extreme cases of disorganization in unfocused spindles that miscategorized them 3.77% of the time, for example where a k-fiber was 'over-aligned' but both of its ends were on the far side of the metaphase plate. Manually correcting these rare cases yielded correlation coefficients −0.33 for control (unchanged) and −0.17 for unfocused (compared to −0.18 reported in the figure).

## Tracking photobleach marks along k-fibers (Figures 3 and 4)

Spindles of k-fibers with photobleach marks were registered by the tub-GFP channel to account for global spindle translations and rotations. Videos of ablated k-fibers were not registered due to expected translocation of k-fibers stubs after ablation. All videos were trimmed to be isochronous, then flipped, rotated, and cropped so that individual k-fibers with photomarks were latitudinal, with chromosomes on the left and minus-ends on the right. A line with width 5 pixels was drawn along individual k-fibers, and the max-intensity projection along the height at each time point was plotted to generate kymographs. Segmented lines were drawn along the kymographs corresponding to the positions of the kinetochore, photomark, and minus-end or pole over time. The distance between the mark and the minus-end over time was calculated and plotted in Python.

## Cell division analysis (Figure 5)

Quantifications of cell division were performed in FIJI. Chromatid separation was quantified by tracking distance between sister chromatids, specifically between the plus-ends of their attached k-fibers, starting the frame before chromatid separation was first observed and ending at the onset of cytokinesis marked by the appearance of a cleavage furrow. To quantify the fraction of chromosome mass per daughter 'cell', 'cell' outlines were drawn based on phase contrast images, and the overlap of each cell outline with the summed intensity z-projection of chromosome masses was measured.

## Statistical analysis

Statistical analyses were performed in Python using NumPy and SciPy unless otherwise stated. Linear regression and Pearson's correlation coefficient calculations were performed using SciPy. In the text, whenever we state a significant change or difference, the p-value for those comparisons was less than 0.05. In figures, *$p < 0.05$, **$p < 0.005$, and ***$p < 0.0005$. In the figure legends, we display the exact p-value from every statistical test made. We used a two-tailed Welch's *t*-test everywhere unless otherwise stated, since this compares two independent datasets with different standard deviations. Legends include *n*, the number of individual measurements made, and *N*, the number of unique cells assayed for each condition.

## Autocorrelation (Figure 3A)

Autocorrelation analysis was performed using Wolfram Mathematica 13.0. The autocorrelation is calculated by the built-in function 'CorrelationFunction'. By this definition, the autocorrelation of a k-fiber at lag $h$ is $\sum_{i=1}^{n-h}(x_i - \bar{x})(x_{i+h} - \bar{x})/\sum_{i=1}^{n}(x_i - \bar{x})^2$ where $x_i$ is k-fiber length at time $i$ and $\bar{x}$ is the mean

of $x_i$. The standard deviation is calculated by the built-in function 'StandardDeviation'. Statistical significance was performed using the built-in function 'LocationTest' at each $h$.

## Script packages

All scripts were written in Python using Spyder through Anaconda unless otherwise stated. Pandas was used for data organization, SciPy for statistical analyses, Matplotlib and seaborn for plotting and data visualization, SciKit for image analysis, and NumPy for general use. FIJI was used for video formatting, intensity quantification, kymograph generation, and tracking k-fibers.

## Video preparation

Videos show a single spinning disk confocal z-slice imaged over time (*Figure 2—video 1*, *Figure 2—video 2*, *Figure 3—video 1*, *Figure 4—video 1*, *Figure 4—video 2*, *Figure 5—video 1*) or a maximum intensity projection (*Figure 1—video 1*, *Figure 1—video 2*, *Figure 1—video 3*) and were formatted for publication using FIJI and set to play at 10 fps.

## Acknowledgements

We thank Tim Mitchison for PtK2 cells, Mollie Meffert for the p50 construct, Dan Needleman, Trina Schroer, Wallace Marshall, Orion Weiner, David Agard, Fred Chang, and Christina Hueschen for helpful discussions, and Arthur Molines, Alex Long, Miquel Rosas Salvans, and other members of the Dumont Lab for discussions and critical reading of the manuscript. This work was supported by NIHR35GM136420, NSF CAREER 1554139, NSF 1548297 Center for Cellular Construction, Chan Zuckerberg Biohub, UCSF Byers Award, UCSF PBBR Award (SD); NSF Graduate Research Fellowship (MR), ARCS Foundation, BISHOP (MR); Fannie and John Hertz Foundation Fellowship (LN); American Heart Association Predoctoral Fellowship (NC); and UCSF Discovery Fellows Program (LN, NC).

## Additional information

### Funding

| Funder | Grant reference number | Author |
|---|---|---|
| Achievement Rewards for College Scientists Foundation | Graduate Student Scholarship | Manuela Richter |
| National Science Foundation | Graduate Research Fellowship Program | Manuela Richter |
| University of California, San Francisco | PIBS Bishop Fellowship | Manuela Richter |
| National Institutes of Health | NIHR35GM136420 | Sophie Dumont |
| National Science Foundation | NSF CAREER 1554139 | Sophie Dumont |
| National Science Foundation | NSF 1548297 Center for Cellular Construction | Sophie Dumont |
| Chan Zuckerberg Initiative | Chan Zuckerberg Biohub | Sophie Dumont |
| University of California, San Francisco | Byers Award | Sophie Dumont |
| University of California, San Francisco | Program for Breakthrough Biomedical Research (PBBR) | Sophie Dumont |
| Hertz Foundation | Hertz Fellowship | Lila Neahring |
| American Heart Association | Predoctoral Fellowship | Nathan H Cho |

| Funder | Grant reference number | Author |
| --- | --- | --- |
| University of California, San Francisco | Discovery Fellows Program | Lila Neahring Nathan H Cho |

The funders had no role in study design, data collection, and interpretation, or the decision to submit the work for publication.

## Author contributions

Manuela Richter, Conceptualization, Resources, Data curation, Software, Formal analysis, Supervision, Funding acquisition, Validation, Investigation, Visualization, Methodology, Writing - original draft, Writing - review and editing; Lila Neahring, Conceptualization, Resources, Data curation, Supervision, Funding acquisition, Methodology, Project administration, Writing - review and editing; Jinghui Tao, Conceptualization, Data curation, Software, Formal analysis, Funding acquisition, Methodology, Writing - review and editing; Renaldo Sutanto, Software, Formal analysis, Validation; Nathan H Cho, Resources, Formal analysis, Validation, Writing - review and editing; Sophie Dumont, Conceptualization, Resources, Supervision, Funding acquisition, Methodology, Project administration, Writing - review and editing

## Author ORCIDs

Manuela Richter ⓘ http://orcid.org/0000-0002-9366-4038
Lila Neahring ⓘ http://orcid.org/0000-0003-2272-8732
Renaldo Sutanto ⓘ http://orcid.org/0000-0002-1252-1482
Nathan H Cho ⓘ http://orcid.org/0000-0002-0110-1343
Sophie Dumont ⓘ http://orcid.org/0000-0002-8283-1523

## Decision letter and Author response

Decision letter https://doi.org/10.7554/eLife.85208.sa1
Author response https://doi.org/10.7554/eLife.85208.sa2

# Additional files

## Supplementary files

• MDAR checklist

## Data availability

We provide all source data, both raw and analyzed, used to generate plots for each figure. In the source files, data were assigned a unique identifier with nomenclature (e.g. '211217_001_01b_MAX13-19'). Characters 1–10 correspond to the unique cell identifier (e.g. '211217_001'), characters 12–14 correspond to the unique k-fiber and its spindle half denoted by 'a' or 'b' (e.g. '01b'), and the final characters correspond to the z-planes over which the k-fiber spanned, for example '13–19'. We provide source code for *Figure 1*.

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
