## [Editor Report]

The authors find that the control of overall kinetochore fiber length in mitotic spindles of cultured mammalian cells does not require spindle pole focusing. Pole focusing is however required for fine-tuning their length and for correct chromosome segregation. Technically sophisticated experiments and careful quantitative data analysis provide compelling evidence for the drawn conclusions that provide valuable insight into spindle morphogenesis. This work is of particular interest to cell biologists and biophysicists interested in cytoskeleton organization.

---

## [Decision Letter]

**Decision letter after peer review:**

Thank you for submitting your article "Kinetochore-fiber lengths are maintained locally but coordinated globally by poles in the mammalian spindle" for consideration by eLife. Your article has been reviewed by 2 peer reviewers, one of whom is a member of our Board of Reviewing Editors, and the evaluation has been overseen by Anna Akhmanova as the Senior Editor. The following individual involved in the review of your submission has agreed to reveal their identity: Jesse C Gatlin (Reviewer #2).

Essential revisions:

1) Please clarify to which extent dynein is inhibited and put the observed phenotype studied here into context with prior work (points 1-3 of reviewer 1).

2) Please clarify to which extent the position and the length of severed k-fibers may affect the observed response in order to disentangle a possible influence of these parameters from the effect of pole focusing versus no pole focusing (points 1-2 of reviewer 2).

3) Please consider improving the quality of some of the data presented in Fig. 4 (point 5 of reviewer 1 and point 3 of reviewer 2).

---

## [Author Response]

Essential revisions:1) Please clarify to which extent dynein is inhibited and put the observed phenotype studied here into context with prior work (points 1-3 of reviewer 1).

We thank the reviewers for these comments and agree that characterization of dynein inhibition and better context would strengthen the manuscript. To address these points, we have added a discussion of context to the manuscript and added Figure 1—figure supplement 1 and Figure 1—figure supplement 3 to characterize our method of dynein inhibition.

We believe that p50 overexpression is similar to other dynein inhibition methods that disrupt poles in mammalian cells (which include dynein depletion, dynein antibody injection, and NuMA depletion) whereby poles unfocus and k-fiber are disorganized (Gaglio et al., 1997; Hueschen et al., 2017; Silk et al., 2009). At the same imaging timescale, the p50-unfocused spindles we observed were similar to previously reported NuMA-KO or dynein-KO spindles; over hours, p50-overexpressed spindles exhibited some turbulence (Figure 1A, Figure 1—video 2) similar to NuMA-KO and dynein-KO spindles (Hueschen et al., 2019). On the shorter, minutes timescales of this work, p50-overexpressed spindles exhibited steady-state phenotypes, where k-fibers seem to remain largely in place without massive rearrangements (Figure 1—video 3). We now clarify the relevant timescale of observation in line 137: “…here we selected unfocused spindles that maintained a steady-state structure on the minutes timescale, holding their shape over time (Figure 1—video 3).”

To provide better context, we have added the following in line 129 of Results: “p50 overexpression dissociates the dynactin complex and inhibits the pole-focusing forces of its binding partner, dynein (Echeverri et al., 1996; Howell et al., 2001; Quintyne et al., 1999), unfocusing poles in species such as *Xenopus* and *Drosophila* (Gaetz and Kapoor, 2004; Sharp et al., 2000). Unfocused spindles correlated with higher mean intensity levels of p50 expression (see newly added Figure 1—figure supplement 1), consistent with prior work showing mild pole disruption versus severe unfocusing depending on the severity of dynein inhibition (Elting et al., 2017; Gaglio et al., 1997, p. 199; Hueschen et al., 2019, 2017; Sharp et al., 2000, p. 200; Toorn et al., 2022). Additionally, we overexpressed p50 in RPE1 cells, showing reproducibility of unfocused spindle phenotypes across other cell types (see newly added Figure 1—figure supplement 3).

All together, we believe that the discussion and characterization of dynein inhibition phenotypes and the newly added figures will provide better context for readers to both understand our work and relate it to previous and future work.

2) Please clarify to which extent the position and the length of severed k-fibers may affect the observed response in order to disentangle a possible influence of these parameters from the effect of pole focusing versus no pole focusing (points 1-2 of reviewer 2).

We thank the reviewers for the insightful comments on correlating position and length of k-fibers and agree this analysis would inform on potential length-setting mechanisms. To address these comments, we have performed spatial correlation analyses along the spindle major and minor axes as suggested (see newly added Figure 1—figure supplement 6).

These new data are discussed in line 195: “To test the role of poles in coordinating lengths within the spindle across space, we tested whether k-fiber length correlated with k-fiber spatial positioning within the spindle. Based on geometry and previous observations, we expected outer k-fibers to be longer than inner k-fibers in focused spindles (Kiewisz et al. 2022). This was indeed the case in control spindles, but this difference was lost in unfocused spindles (Figure 1—figure supplement 6A). Furthermore, we expected k-fiber length to correlate with distance to the metaphase plate – k-fibers are shorter if attached to under-aligned chromosomes and longer if attached to over-aligned chromosomes (Wan et al., 2012). Here too, a correlation between k-fiber length and alignment was observed in control but it was negligible in unfocused spindles (Figure 1—figure supplement 6B). Thus, poles coordinate k-fiber lengths spatially in the spindle to maintain its shape despite geometric constraints.” We now also refer to these data in the context of the Ran-GTP model for setting length in line 387: “…while the globally disorganized structure of unfocused spindles (Figure 1B,D) could lead to modified gradients, the mean length of k-fibers is unchanged and length does not correlate with spatial position along both spindle axes (Figure 1E, Figure 1—figure supplement S6).”

The reviewers also raise a discerning point about correlating position and length of ablated k-fibers with dynamics and growth rate. We are also keenly interested in probing whether regrowth dynamics are length-dependent; however, these data are noisy due to ablated k-fibers minus-ends moving in and out of z-plane during early length recovery (when they are poorly anchored in the spindle) in control and unfocused spindles, as well as due to high microtubule density near focused poles obscuring minus-ends during late length recovery. We have not observed any statistically significant trends, and we are not comfortable making any conclusions about the possibility of length-dependent growth rates.

3) Please consider improving the quality of some of the data presented in Fig. 4 (point 5 of reviewer 1 and point 3 of reviewer 2).

We thank the reviewers for their close examination of the data and recommendations for clarifying the data. We have reworded the manuscript and adjusted figures to improve clarity, readability, and transparency, as well as added Figure 4—figure supplement 1 to demonstrate the reversible tuning of k-fiber minus-end dynamics after length recovery.

The reviewers suggest probing whether the apparent loss of flux is significant by performing more ablation and photomarking experiments in unfocused spindles, but a power analysis reveals this would not be feasible on a reasonable time scale. To detect a statistically significant decrease in flux after ablation in unfocused k-fibers using our current dataset (flux in unfocused spindles: 0.13 ± 0.15 µm/min versus ablated unfocused k-fibers: -0.03 ± 0.23 µm/min), we would need a sample size of n = 38 for each condition based on a power analysis with 80% confidence and significance 0.05. This is not feasible on a reasonable time scale given the technical difficulty and low throughput nature of the experiment. Since we did not perform additional ablations with photomarks in unfocused spindles, we sought to ensure that the language in the Results and Discussion is clear about what we can conclude and what we speculate.

We discuss this lack of statistical significance in line 329 of the Results: “In unfocused spindles, the photomark also did not detectably approach the minus-end of the k-fiber during its regrowth (Figure 4C), consistent with suppression of any minus-end dynamics, though it was not significantly different from the already slow dynamics and insufficient to account for growth (Figure 4D). Thus, k-fibers can tune their plus-end dynamics to recover their length in the absence of dynein-based pole-focusing forces.” For this reason, in the Discussion line 409, we state, “Our findings suggest that in response to length changes, k-fibers regulate their plus-end dynamics in an analog manner and their minus-end dynamics in a digital manner. In unfocused spindles, we have shown that the regrowth of shortened k-fibers is driven by an increase in plus-end polymerization, and that this occurs in response to length changes, not simply dynein-based force changes (Figure 4).” Thus, we propose a digital mechanism for how k-fibers regulate their minus-end dynamics without concluding this mechanism is independent of pole-focusing, although our data are consistent with it. While the original manuscript was written consistently with what we know, we agree with reviewers that these conclusions can be made more transparent, so we have added the following to line 415 of the Discussion: “In turn, after ablation, we always observed a near-absence of minus-end dynamics. This reduction in flux was only large enough in control k-fibers to observe statistical significance, though unfocused k-fibers appeared to follow the same trend (Figure 4C, D).”

To address the quality of the assay and bolster our conclusions, we performed additional photomarking in control spindles (see newly added Figure 4—figure supplement 1). These data are discussed in newly added line 322 in the Results: “Consistent with k-fiber minus-end dynamics being transiently suppressed during regrowth, rather than frozen due to ablation damage, k-fiber minus-ends resumed depolymerization in control spindles after ablation and length recovery (Figure 4—figure supplement 1, Figure 4—video 2).” These results, combined with previous work, indicate that ablation does not lead to dysfunctional k-fiber minus-ends since k-fiber length, biochemistry, and dynamics are restored. These data further support that minus-end dynamics are indeed tunable in control k-fibers, since minus-end depolymerization is not detected during length recovery immediately following ablation, but it is detected at near-baseline flux rates after length recovery. We speculate that a similar mechanism occurs in unfocused k-fibers, but the difference in baseline flux and flux during length recovery was too small to see statistically significant trends.

Finally, we improved the presentation of figures and annotations in Figures 2, 3, and 4. We added the same markers in the time strips and kymographs as used in the videos to clearly show features of interest and how they move over time, as well as removed extraneous annotations. We also added cartoons and labeled, color-coordinated lines to guide the reader’s eyes along these features in kymographs.

We thank the reviewers for their critical consideration of experiments and conclusions, and we believe that the language and figures show our results transparently and logically. We also believe the additional experiments confirm that k-fibers are not permanently damaged by ablation, and that focused k-fibers are indeed able to dynamically pause and restart minus-end depolymerization in response to length changes.